# Role of 5-HT2A, 5-HT2C, 5-HT1A and TAAR1 Receptors in the Head Twitch Response Induced by 5-Hydroxytryptophan and Psilocybin: Translational Implications

**DOI:** 10.3390/ijms232214148

**Published:** 2022-11-16

**Authors:** Orr Shahar, Alexander Botvinnik, Noam Esh-Zuntz, Michal Brownstien, Rachel Wolf, Amit Lotan, Gilly Wolf, Bernard Lerer, Tzuri Lifschytz

**Affiliations:** Biological Psychiatry Laboratory and Hadassah BrainLabs, Hadassah Medical Center, Hebrew University Jerusalem, Jerusalem 91120, Israel

**Keywords:** psychedelics, psilocybin, 5-HTP, head twitch response, psychiatric disorders, serotonin receptors, TAAR1, receptor modulation, C57BL

## Abstract

There is increasing interest in the therapeutic potential of psilocybin. In rodents, the serotonin precursor, 5-hydroxytryptophan (5-HTP) and psilocybin induce a characteristic 5-HT2A receptor (5-HT2AR)-mediated head twitch response (HTR), which is correlated with the human psychedelic trip. We examined the role of other serotonergic receptors and the trace amine -associated receptor 1 (TAAR1) in modulating 5-HTP- and psilocybin-induced HTR. Male C57BL/6J mice (11 weeks, ~30 g) were administered 5-HTP, 50–250 mg/kg i.p., 200 mg/kg i.p. after pretreatment with 5-HT/TAAR1 receptor modulators, psilocybin 0.1–25.6 mg/kg i.p. or 4.4 mg/kg i.p., immediately preceded by 5-HT/TAAR1 receptor modulators. HTR was assessed in a custom-built magnetometer. 5-HTP and psilocybin induced a dose-dependent increase in the frequency of HTR over 20 min with attenuation by the 5-HT2AR antagonist, M100907, and the 5-HT1AR agonist, 8-OH-DPAT. The 5-HT2CR antagonist, RS-102221, enhanced HTR at lower doses but reduced it at higher doses. The TAAR1 antagonist, EPPTB, reduced 5-HTP- but not psilocybin-induced HTR. We have confirmed the key role of 5-HT2AR in HTR, an inhibitory effect of 5-HT1AR, a bimodal contribution of 5-HT2CR and a role of TAAR1 in modulating HTR induced by 5-HTP. Compounds that modulate psychedelic-induced HTR have important potential in the emerging therapeutic use of these compounds.

## 1. Introduction

Psychiatric disorders cause significant individual suffering as well as have a major economic impact [1,2] and manifest a high level of resistance to standard treatment [3,4]. Novel therapeutic approaches are needed. Recent findings suggest that psychedelic compounds may fill this role. Psychedelics significantly alter consciousness, cognition and mood [5,6] and have been used for centuries for healing and spiritual purposes [7,8].

After Albert Hoffman synthesized lysergic acid diethylamide (LSD) in 1938 and discovered its psychedelic properties, he synthesized psilocybin, which is the principal psychoactive component of “magic mushrooms” [9]. A significant body of mostly uncontrolled research followed but was halted when psychedelic drugs were declared Schedule 1 substances in 1968 [6]. After decades of relative inactivity, research on psychedelic compounds has increased significantly, including controlled studies in depression [10,11] and posttraumatic stress disorder [12,13]. Several trials are ongoing [14]. The principal emphasis has been on serotonergic psychedelics which include tryptamines such as psilocybin, N,N-dimethyltryptamine (DMT) and 5-methoxy-N,N-dimethyltryptamine (5-MeO-DMT); phenethylamines such as mescaline; and ergolines such as LSD.

Serotonergic psychedelic agents such as psilocybin are thought to act mainly via the serotonin 5-HT2A receptor to which psilocybin’s active metabolite, psilocin, binds with high affinity [15,16]. Although it is known that serotonergic psychedelics bind to an array of serotonin receptors including but not limited to 5-HT1A/2A/2B/2C [17], it is not fully understood how activation of different serotonin receptors interacts to exert the subjective and neurobiological effects of psychedelics [18].

In rodents, psilocybin induces a characteristic head twitch response (HTR), which is correlated with the psychedelic trip in humans and across rodent species [19]. Because of increasing interest in the development of novel psychedelic analogues, including variants that do not induce the characteristic “trip” [20,21], HTR is an important preclinical tool in the evaluation of psychedelic compounds. HTR was first described in mice after administration of the serotonin precursor, 5-hydroxytryptophan (5-HTP) [22], and has been further characterized by subsequent investigators [23,24,25,26,27,28,29]. Although extensive research has documented the effect of 5-HTP to induce HTR in rodents [30,31,32,33], psychedelic effects have not been reported at doses administered to humans [34].

Studies in humans have shown that 5-HT2A antagonists, such as ketanserin can block the short-term subjective effects of psilocybin and other serotonergic psychedelics [35,36]. Attenuation of subjective psychedelic effects has also been reported for the 5-HT1A partial agonist, buspirone [37]. It is well-established that 5-HT2A antagonists block HTR in mice administered psilocybin and other serotonergic psychedelics [38]. Fantegrossi, Simoneau, Cohen, Zimmerman, Henson, Rice and Woods [25] have shown that the 5-HT2C antagonist, RS102221, enhances DOI-elicited-HTR in C57Bl/6 mice at doses up to 3 mg/kg and attenuates HTR at a dose of 10 mg/kg. Custodio et al. [39] showed substantial attenuation of HTR induced by DOI and two derivatives of mescaline by pre-treatment with the 5-HT2C antagonist, SB-242084. A very interesting finding was that one of the mescaline derivatives induced HTR that was blocked by the 5-HT2C antagonist, SB242084, but not by a 5-HT2A antagonist (ketanserin), suggesting that the HTR-inducing effect was mediated by 5-HT2CR, independent of 5-HT2AR. [39].

In the current study, we sought to define the dose–response characteristics of HTR induced by 5-HTP and chemically synthesized psilocybin (PSIL) in C57Bl/6j mice to examine the role of 5-HT2A, 5-HT2C and 5-HT1A receptors in modulating HTR induced by 5-HTP and PSIL using appropriate agonists and antagonists and to determine a possible role of the trace amine associated receptor 1 (TAAR1) in mediating HTR induced by 5-HTP and psilocybin. Our findings confirm the pivotal role of 5-HT2A receptors in HTR induced by 5-HTP and PSIL, support an inhibitory role of 5-HT1A receptors and an inverted U dose–response effect of a 5-HT2C receptor antagonist and suggest a role for TAAR1 receptors in the HTR-inducing effect of 5-HTP.

## 2. Results

### 2.1. Effect of 5-HTP on HTR

Initially, we evaluated the effect of increasing doses (50 mg/kg to 250 mg/kg) of 5-HTP on HTR. These 5-HTP doses induced a dose-dependent increase in HTR (Figure 1A) reflected in a between-subject main effect of dose in a two-way repeated measure ANOVA. HTR rate increased during the test, indicated by within-subject effect of time and by time x dose interaction (See Appendix A for detailed results of statistical analyses). When comparing total HTR (Figure 1B), there was a notable increase in HTR at the higher doses. A further analysis of the dose–response results was conducted into a pharmacological dose–response logarithmic curve (Figure 1C), and the data that were included were taken from the 5-HTP induction of Total HTR (Figure 1B). 5-HTP induced the greatest HTR (Figure 1A) during the last 15 min of the 30 min measurement. We therefore decided it would be most relevant to analyze the effects of receptor modulators on 5-HTP-induced HTR during the 15–30 min measurement period for the following experiments.

To examine the effects of serotonin and TAAR1 receptor modulators, 5-HTP 200 mg/kg was administered following a dose of the modulators (Figure 2). The 5-HT2A receptor antagonist, M100907 (volinanserin) (Figure 2A), completely blocked 5-HTP-induced HTR at both 0.5 mg/kg and 2 mg/kg doses. Pre-treatment with the 5-HT1A receptor agonist, 8-OH-DPAT 1 mg/kg and 2 mg/kg (Figure 2B), significantly reduced 5-HTP-induced HTR. The 5-HT2C antagonist, RS-102221 (Figure 2C), did not alter 5-HTP-induced HTR at 2 mg/kg, and significantly increased 5-HTP-induced HTR at 4 mg/kg and 8 mg/kg. The TAAR1 antagonist, EPPTB (Figure 2D), significantly reduced 5-HTP-induced HTR at both doses of 1 mg/kg and 10 mg/kg. Analyzing 15–30 min (Figure 2) compared to the full time course of 30 min can indicate some differences in the total HTR between Figure 1B and Figure 2.

### 2.2. Effect of Psilocybin on HTR

First, we assessed the effect of increasing PSIL concentrations (0.1 mg/kg to 25.6 mg/kg) on mouse HTR. HTR over time (2 min time bins) showed a dose-dependent increase in head twitch up to a dose of 25.6 mg/kg (Figure 3A) reflected in a between-subject main effect of dose in a two-way repeated measure ANOVA (see Appendix A for detailed results of the statistical analyses). HTR rate increased during the test, indicated by within-subject effect of time and by time x dose interaction. The dose–response graph was observed to manifest a bimodal profile (Figure 3B).

The lower PSIL doses from 0 to 1.6 mg/kg produced a gradual increase followed by a slightly sustained HTR and then a gradual decrease in the effect when compared to the higher doses from 3–25.6 mg/kg, which produced a very sharp onset with a fast decline of HTR without a sustained period. The doses were compiled into two groups (0.1–1.6 mg/kg and 3–25.6 mg/kg) and analyzed. Total PSIL-induced HTR during the 20 min measuring period was compiled (Figure 3C); the doses that induced the most HTR over time when compared to vehicle were 1.6 mg/kg, 3.2 mg/kg, 4.4mg/kg, 6.4 mg/kg and 25.6 mg/kg. An important aspect of the evaluation was to examine the maximal effect that PSIL had on inducing HTR (Figure 3D), which was determined through peak HTR comparison. When compared to vehicle, a significant difference was shown for the higher doses (above 1.6 mg/kg) in a similar fashion to the bimodal pattern seen in Figure 3A. A further analysis of the dose–response results was conducted into a pharmacological dose–response logarithmic curve (Figure 3E), and the data points that were included were taken from PSIL induction of maximal peak HTR (Figure 3D).

For human clinical trials, 25 mg/70 kg is administered as a representative high psilocybin dose [40,41]. Using DoseCal, a virtual calculator for dosage conversion between human and different animal species [42], a 25 mg/70 kg human dose was converted to 4.4 mg/kg in mice. The administration of 4.4 mg/kg PSIL stimulated a strong acute HTR response (Figure 3A). Although this dose was administered i.p. and is thus not analogous to an oral dose in humans, it was chosen as a representative acute PSIL dose for further receptor modulator experiments.

To examine the effects of serotonin and TAAR1 receptor modulators, PSIL 4.4 mg/kg was administered following doses of the modulators (Figure 4). 20 min post injection of PSIL, M100907 and 8-OH-DPAT (Figure 4A) completely blocked HTR at all doses. Twenty minutes post injection of PSIL, treatment with RS-102221 (Figure 4B) did not alter PSIL-induced HTR with 2 mg/kg, significantly increased PSI-induced HTR with 4 mg/kg and reduced PSIL-induced HTR with 8 mg/kg. EPPTB (Figure 4C) did not alter (no statistical sig.) PSIL-induced HTR with both doses of 1 mg/kg and 10 mg/kg.

To assess any difference between the 5-HTP and PSIL induction profiles of HTR, we chose doses with comparable maximal HTR. 5-HTP 200 mg/kg was compared to PSIL 3 mg/kg (Figure 5). A somewhat “mirror image” HTR profile was obtained. Although both doses of compounds reached the same peak HTR, the way the HTR was induced was different, with the PSIL effect peaking earlier in the time course and the 5-HTP effect peaking later.

Detailed statistical results for all statistics represented in all figures can be seen in Appendix A.

## 3. Discussion

The present study sought to ascertain the effect of 5-HTP and PSIL on HTR over a range of doses and to further our understanding of the receptor mechanisms implicated by pre-treatment administration of receptor modulators. For 5-HTP, we defined 200 mg/kg as the highest HTR-inducing dose and demonstrated that both the 5-HT2A antagonist, M100907, and the 5-HT1A agonist, 8-OH-DPAT, significantly blocked the effect. The 5-HT2C antagonist, RS-102221, increased 5-HTP-induced HTR up to 4 mg/kg; a reduction in HTR was observed with 8 mg/kg. The TAAR1 antagonist EPPTB caused a reduction in 5-HTP-induced HTR with an effective dose of 1 mg/kg. For PSIL, we observed a bimodal profile of HTR, whereby doses from 1.6 mg/kg elicited a rapid increase in HTR that was followed by a rapid decrease. The higher the administered dose, the more profoundly this effect was demonstrated. Pre-treatment with the 5-HT2A antagonist, M100907, and the 5-HT1A agonist, 8-OH-DPAT, significantly blocked HTR induced by PSIL 4.4 mg/kg. The 5-HT2C antagonist, RS-102221, increased PSIL-induced HTR up to 4 mg/kg; a significant reduction in HTR was seen with 8 mg/kg. Unlike 5-HTP, there was no effect of the TAAR1 antagonist, EPPTB, on PSIL-induced HTR.

HTR is characterized by a fast, vigorous burst of left to right head-shake movements [27,28]. Before the development of automated detection methods, manual scoring of HTR was performed, a labor-intensive process that is conducted on one mouse at a time by one scorer. Evolution in the HTR recording field has brought about the use of the magnetometer in conjunction with magnets implanted on the skull of the animals [43]. In the current study, we chose to avoid the surgical procedure for implanting a magnet on the animal’s skull and used the novel procedure of attaching magnetic ear tags [44]. The magnetic ear tags are well-tolerated in mice that are housed in groups. Possible disadvantages can include inflammatory processes that can develop over longer periods of time (during two months after the tagging) and limitations relating to the age and size of the animals. A major advantage is the ability to measure multiple animals as opposed to one when performed manually. For the current study, a set-up of six individual coiled-containers was used. When using magnetometer recording of HTR, the software used can distinguish between the frequency induced by HTR and the frequency associated with other head movements [45].

An alternative, recently described tool for measuring HTRs in a non-invasive way is the TopScan (Clever Sys Inc, Reston, VA, USA) computer software-based scoring [46] system. This system is limited to one mouse per recording box. Another option that was implemented in a recent study is the use of a high-resolution camera recording paired with the software analysis DeepLabCut pro (Mathis Laboratory), which gives video tracking evaluation from a recording but is limited to the number of recordings one can generate as well as false negatives that may arise from the positioning of the mouse during an HTR which might be hidden from the camera and not picked up [47].

5-HTP-induced HTR has previously described by multiple authors [30,31,32,33,48]. However, 5-HTP has not been reported to have psychedelic effects in humans [49]. Although, overdoses of compounds that increase serotonin release can result in serotonin syndrome, which may include hallucinations [50,51], classic psychedelic effects resembling those induced by tryptaminergic and other psychedelic drugs have not been reported. In our study, administration of 5-HTP at 150–250 mg/kg induced significant HTR. The implications of administering equivalent high doses of 5-HTP to humans are unknown. There are two instances of administering up to 3000 mg 5-HTP per os per day but not as a single dose. Such prolonged exposure that can result in tolerance effects [49].

Contrasting the induction of HTR between 5-HTP and PSIL, a comparison of peak HTR inducing doses of both drugs revealed interesting insights. Firstly, 5-HTP-induced HTR reached a peak at 28 min whereas PSIL induced HTR reached a peak at 4 min. Second, the sharp onset of PSIL that reaches peak HTR during the first 4 min of measurement is associated with a rapid decline, reaching half maximal HTR by 11 min. On the other hand, 5-HTP-induced HTR gradually increased over 28 min.

We have categorized the effect on HTR induction by 5-HTP and PSIL of pre-treatment with 5-HT2A and 5-HT2C antagonists and a 5-HT1A agonist. We found complete ablation of HTR induced by 5-HTP and PSIL following pre-treatment with the 5-HT2A antagonist M100907, similar to the effect of M100907 on HTR induced by DOI [44,52], 2-CI [53], N,N-dipropyltryptamine (DPT) [54] and LSD [55,56,57]. The 5-HT1A agonist, 8-OH-DPAT, significantly attenuated HTR induced by 5-HTP and PSIL. Not much literature is available on the effect of 8-OH-DPAT and other 5-HT1A agonists on HTR induced by psychedelic drugs; nevertheless, it has been shown to inhibit DOI-induced head twitch behavior in naive rats [58,59]. In the current study, we observed that the 5-HT2C antagonist RS-102221 generated an increase in HTR induced by 5-HTP and PSIL up to a dose of 4 mg/kg and decreased HTR with 8 mg/kg. Our observation is comparable to a study that was conducted on the effects of RS-102221 on DOI-induced HTR [25]. Furthermore, DOI-elicited HTR is reduced in 5-HT2C receptor knockout mice [60], suggesting the important role that 5-HT2C has in mediating HTR and potentially eliciting psychedelic-like subjective effects in humans. Another recent study found that one of the derivatives of DOI (methallylescaline) produced HTR that was blocked by 5-HT2C antagonist (SB-242084) but not by a 5-HT2A antagonist (ketanserin), indicating that HTR-inducing effect might be mediated by 5-HT2C independently from 5-HT2A [39].

Interestingly, the TAAR1 antagonist EPPTB decreased HTR induced by 5-HTP while not affecting PSIL-induced HTR. There are studies conducted on a broad range of psychoactive drugs regarding their interaction with TAAR1 [61,62]. EPPTB (5 mg/kg) prevented the inhibitory effect of LSD (30–150 µg/kg) on VTA dopamine firing activity [63]. Although some psychedelics have been found to interact with TAAR1, it seems that there are clear differences between pharmacologically close substances [64]; for example, it has been shown that relative to DMT, 5-MeO-DMT generated a substantially lower TAAR1-induced cAMP response, indicating a lower effectiveness at this receptor [62]. In the same study, it has been shown that 5-HTP stimulated a substantial cAMP response at TAAR1 [62]. Current research regarding psilocybin interaction on TAAR1 is interesting; Simmler, Buchy, Chaboz, Hoener and Liechti [61] have shown that psilocin has Ki values of 1.4 and 17, in rat and mouse TAAR1, respectively. Further studies are needed to clarify the role of TAAR1 receptors in the action of psychedelic drugs.

Our findings should be considered in the context of a recent paper by Erkizia-Santamaría et al. [65], who examined the effect of psilocybin 0.125–3.0 mg/kg i/p on manually rated HTR and the modulatory effects of co-administrating psilocybin with 5-HT2A, 5-HT2C and 5-HT1A serotonin receptors antagonists. The most important difference between the findings of the two studies is in the dose–response results where we found that the maximal HTR effect was reached with 25.6 mg/kg compared to Erkizia-Santamaria et al.’s result of 1 mg/kg. It is noteworthy that Erkizia-Santamaría et al. [65] measured HTR by a human observer and did so only between 5–25 min in 5 min bins. We have shown that for psilocybin doses, above 1.6 mg/kg, most of the HTR response lies between 0 and 6 min, where the most profound effect can be seen (Figure 3A). The difference between the studies is apparent when comparing total HTR figures, where Erkizia-Santamaría et al. [65] show 10 total HTR for 3mg/kg, whereas we showed around 50 total HTR; both groups measured 20 min of HTR but in different time window windows (we measured 0–20 min whereas Erkizia-Santamaría et al. [65] measured 5–25 min).

In relation to the modulators tested, both groups investigated a 5HT2A antagonist and a 5HT2C antagonist; however, different modulators were used. While both groups showed a clear effect of the 5-HT2A antagonist to block HTR, Erkizia-Santamaria et al. [65] showed that a 5-HT2C antagonist increased HTR with psilocybin 3 mg/kg (only 1 modulator dose tested), whereas we observed an inverted U-shaped effect with psilocybin 4.4 mg/kg with an increasing dose of 5-HT2C antagonist. A further difference of note is that Erkizia-Santamaria [65] et al. studied the effect of a 5-HT1A antagonist and found no effect, whereas we studied a 5-HT1A agonist and found significant inhibition of HTR which is an important finding and a major difference between the papers. Further differences of note between the papers are that we studied the effect of 5-HTP in addition to psilocybin and showed critical modulation of 5-HTP-induced HTR with 5-HT2A and 5-HT2C antagonists and, these being novel findings, with a 5HT1A agonist and a TAAR1 antagonist.

Automation of HTR measurement is a further important differential characteristic of the two papers. The magnetometer can detect very fast head twitches that can be otherwise missed if a human observer is not extremely focused on the mouse’s head; counting a high number of head twitches is very fatiguing on a human and can reduce the accuracy of manual scoring.

In generalizing the results, we have reported, it should be noted that the current study was performed on male C57BL/6j mice. A recent study [66] found HTR differences across sexes in this mouse strain. Further studies are required to validate the results we have obtained with female mice.

A significant discussion is under way in the resurgent psychedelics field as to whether short-term subjective effects are needed for the therapeutic potential of psychedelics to be achieved [67]. The use of MDMA in the context of psychedelic-assisted psychotherapy has been shown to achieve significant therapeutic effects [13]. In the controlled trial of Carhart-Harris [11] in which psilocybin was shown to be equivalent in efficacy to the SSRI escitalopram, psychedelic doses of psilocybin were used. Nevertheless, the potential to treat psychiatric disorders without the logistics of trip mediation (use of trained professionals, highly specific space and duration of effect) has immense economic value. A recent study showed that the attenuation of depression-like features in mice by administration of psilocybin is independent of 5-HT2A receptor activation (demonstrated by co-administrating the 5-HT2A antagonist ketanserin) [38]. Our findings suggest the potential use of 5-HT1A agonists to decrease HTR. How such attenuation will impact the therapeutic effect of psychedelics remains to be evaluated. In this context, we have recently demonstrated that co-administration of the 5-HT1A partial agonist, buspirone, which attenuated HTR, did not impede the effect of psilocybin to reduce marble burying (a screening test for anti-obsessional effects) in male ICR mice (Singh et al., submitted). More research is needed to determine whether attenuation of HTR in mice and of the psychedelic trip in humans by 5-HT receptor modulators, will impede the potential use of psychedelics to treat psychiatric disorders.

## 4. Materials and Methods

### 4.1. Animals

Experiments were performed on adult (9–12 weeks old) C57BL/6J male mice. Animals were housed under standardized conditions with a 12 h light/dark cycle, stable temperature (22 ± 1 °C), controlled humidity (55 ± 10%), free access to mice colony chow, water and up to 8 per cage. Both male and female experimenters performed the studies and handled the mice. Experiments were conducted in accordance with AAALAC guidelines and were approved by the Authority for Biological and Biomedical Models, Hebrew University of Jerusalem, Israel, Animal Care and Use Committee number: MD-21-16563-4. All efforts were made to minimize animal suffering and the number of animals used. To avoid an effect of tolerance to the drugs, only vehicle and lower dose groups were reused with a two-week waiting period and injected with active compounds or vehicle, respectively. Sample sizes were planned evenly beforehand; however, each mouse was inspected before an injection for any signs of physical damage to body (from other mice or a rare occasion of ear inflammation due to ear tags), and some mice were excluded from a treatment group on the day of the experiment. The mice in each cage were randomly selected for different treatments to avoid treating the whole cage with the same treatment.

### 4.2. Materials

PSIL was supplied by Usona Institute, Madison, WI, USA and was determined by AUC at 269.00 nm (UPLC) to contain 98.75% psilocybin and stored in a cool light-sealed safe. 5-HTP (5-Hydroxytryptophan), M100907 ((R)-(+)-α-(2,3-dimethoxyphenyl)-1-[2-(4-fluorophenyl)ethyl]-4-pipidinemethanol), 8-OH-DPAT((±)-8-Hydroxy-2-(dipropylamino)tetralin hydrobromide) and EPPTB (*N*-(3-Ethoxy-phenyl)-4-pyrrolidin-1-yl-3-trifluoromethyl-benzamide) were purchased from Sigma-Aldrich, Tel Aviv, Israel. RS-102221 (8-[5-(2,4-Dimethoxy-5-(4-trifluoromethylphenylsulphonamido)phenyl-5-oxopentyl]-1,3,8-triazaspiro[4.5]decane-2,4-dione hydrochloride) was purchased from Biotest, Kfar Saba, Israel. 5-HTP, M100907 and EPPTB were stored in 4 °C. 8-OH-DPAT and RS102221 were stored in room temperature. PSIL was dissolved in 100% saline (0.9% NaCl) solution. 5-HTP, M100907, 8-OH-DPAT and RS-10222 were dissolved in 5% DMSO + 95% saline (0.9% NaCl) solution. EPPTB was dissolved in 5% ethanol + 5% kolliphor + 90% saline (0.9% NaCl) solution. All solutions were prepared to the appropriate concentration of 10 µL/g for administration (by intraperitoneal (i.p.) injection). Vehicle-treated condition represents injection of the appropriate solvent and the equivalent volume of the drugs administered. Fresh injectable solutions of the compounds were prepared for each experiment.

### 4.3. Ear Tagging

Assessment by electromagnetic generation of the rapid side-to-side headshake that characterizes HTR requires the installation of small magnets in the outer ears of the mice. For this purpose, we utilized small neodymium magnets (N50, 3 mm diameter × 1 mm height, 50 mg), which were attached to the top surface of aluminum ear tags (supplied by Mario de la Fuente Revenga PhD. of Virginia Commonwealth University). Ear tags were placed through the pinna antihelix and laid in the interior of the antihelix, resting on top of the antitragus, leaving the ear canal unobstructed. This procedure was performed by simple restraint and immobilization of the mouse’s head. Signs of ear tissue damage were rare in the form of redness, and the ear tags were well-tolerated in the mice.

### 4.4. Head Twitch Acquisition

Mice were allowed to recover from ear-tagging for 5–7 days prior to testing. The tagged animals were placed inside a magnetometer apparatus (supplied by Mario de la Fuente Revenga PhD. Of Virginia Commonwealth University, Appendix A) consisting of plastic containers (11.6 cm diameter × 13.3cm height) surrounded by a coil (∼500 turns 30 AWG enameled wire), the output of which was amplified (Pyle PP444 phono amplifier) and recorded at 1000 Hz using a NI USB-6001 (National Instruments, Budapest, Hungary) data acquisition system [44]. Recordings were performed using a MATLAB driver (MathWorks, Natick, MA, USA, R2021a version, along with the NI myDAQ support package) with the corresponding National Instruments support package for further processing. A custom MATLAB script was used to record the processed signal, which was presented as graphs showing the change in current as peaks (mAh). A custom graphic user interface created in our laboratory was used to further process the recording into an Excel spreadsheet.

For validation of the magnetometer, a Hero Black 9 GoPro camera (GoPro) was used to record high frame rate (120 frames per sec) overhead videos (1080p resolution). GoPro camera was mounted ∼ 18 cm above the magnetometer container floor, and randomly selected experimental test sessions were recorded for validation from ongoing experiments. Five mice were used for this evaluation across a range of psilocybin doses (PSIL 0.75 mg/kg (*n* = 1), PSIL 2.2 mg/kg (*n =* 2) and PSIL 4.4 mg/kg (*n =* 1). It is not anticipated that the nature of the head twitch response will be different in accordance with dose; therefore, the dose of PSIL was not taken into account in comparing HTR recorded by the magnetometer and HTR recorded manually. Out of 307 magnetometer peaks that were compared across 5 mice, only 1 was found to be a false positive on visual examination (Appendix A). Corroborating this finding across mice, there was a tight correlation between the number of HTR counted by the magnetometer and the number that was validated manually (R^2^ = 0.9996). Taken together, these data confirm the validity of the magnetometer as a reliable measure of the rodent HTR, consistent with previous reports validating this method [44,45,47]. A sample video used for scoring and demonstrating a HTR is available (Appendix A).

Drugs were administered by intraperitoneal (i.p.) injection immediately before placing the weighed animals into the magnetometer. For dose–response, 5-HTP and PSIL were administered at doses of 50 mg/kg to 250 mg/kg and 0.1 mg/kg to 25.6 mg/kg, respectively. For assessing the effects of modulators, all mice were injected with either 5-HTP or PSIL at doses of 200 and 4.4 mg/kg, respectively, immediately preceded by the 5-HT2A receptor antagonist M100907 (0.5 and 2 mg/kg i.p.), the 5-HT1A receptor agonist 8OH-DPAT (1 and 2 mg/kg i.p.), the 5-HT2C receptor antagonist R-S102221 (2, 4 and 8 mg/kg i.p.) or the TAAR 1 antagonist EPPTB (1 and 10 mg/kg i.p.). HTR was measured for 30 min for 5-HTP and 20 min for PSIL in the magnetometer. M100907 and 8-OH-DPAT were administered in the same experimental set with the same vehicle-PSIL group and statistically analyzed together. In this study, dose–response experiments have shown that vehicle–vehicle-treated mice have minimal to no HTR at baseline and on the modulators experiments, a vehicle–vehicle group was decided to not be included in order to preserve the number of animals used and in the case of the specific interest of this experiment’s outcome.

## Figures and Tables

**Figure 1 ijms-23-14148-f001:**
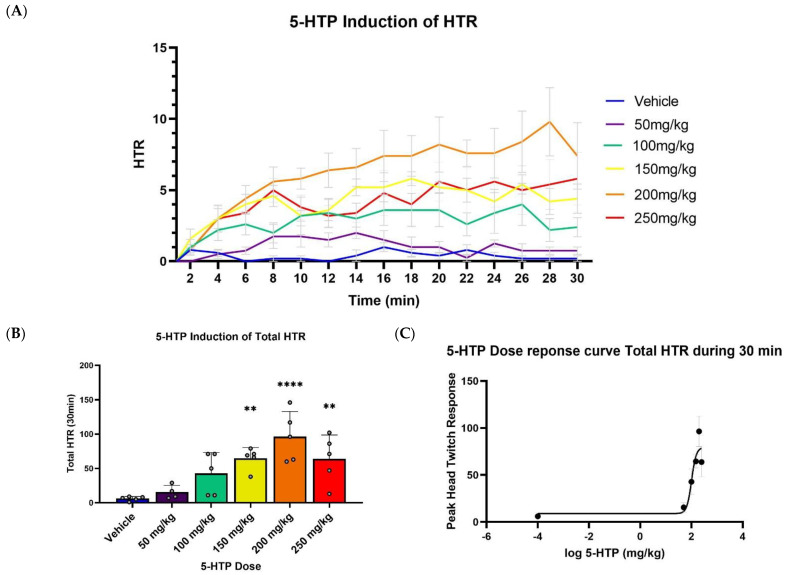
Induction of HTR by 5-HTP, over time (**A**) and total HTR during 30 min (**B**). (**A**) Effect of different 5-HTP concentrations during HTR over the time course of 30 min post-injection (*n =* 4–5). (**B**) Cumulative HTR during the time of 30 min post injection (*n =* 4–5). (**C**) Logarithmic dose–response curve compiled from total HTR (**B**). Compared to vehicle, ** *p* < 0.01, **** *p* < 0.0001. Error bars represent SEM.

**Figure 2 ijms-23-14148-f002:**
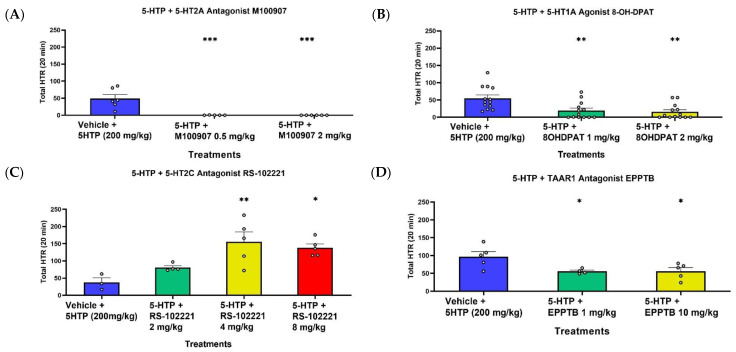
Total 5-HTP-induced HTR (**A**–**D**) during 15–30 min post injection with different co-treatments. (**A**) Effect of pre-treatment with M100907 0.5/2 mg/kg + 5-HTP 200 mg/kg or vehicle + 5-HTP 200 mg/kg (*n =* 5–6). (**B**) Effect of pre-treatment with 8-OH-DPAT 1/2 mg/kg + 5-HTP 200 mg/kg or vehicle + 5-HTP 200mg/kg (*n =* 13). (**C**) Effect of pre-treatment with RS-102221 2/4/8 mg/kg + 5-HTP 200 mg/kg or vehicle + 5-HTP 200 mg/kg (*n =* 3–5). (**D**) Effect of pre-treatment with EPPTB 1/10 mg/kg + 5-HTP 200 mg/kg or vehicle + 5-HTP 200 mg/kg (*n =* 4–5). Compared to vehicle, * *p* < 0.05, ** *p* < 0.01, *** *p* < 0.001. Error bars represent SEM.

**Figure 3 ijms-23-14148-f003:**
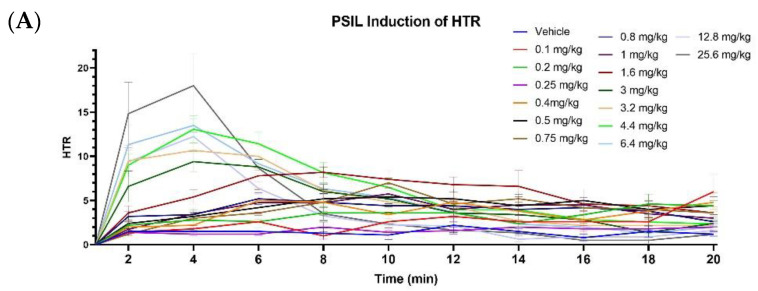
PSIL induced HTR, over time (**A**), bimodal effect (**B**), total HTR during 20 min (**C**) and maximal peak HTR in a 2 min bin over 20 min (**D**) (*n =* 5–14). (**A**) Effect of different PSIL concentrations during HTR over the time course of 20 min post-injection. (B) Bimodal HTR effect of high and low doses over the time course of 20 min post-injection. (**C**) Total HTR during the 20 min post injection. (**D**) Maximal peak HTR that was produced in a 2 min time bin over the course of 20 min (**E**) Logarithmic dose-response curve compiled from peak HTR (**D**). Compared to vehicle, * *p* < 0.05, ** *p* < 0.01, *** *p* < 0.001, **** *p* < 0.0001. Error bars represent SEM.

**Figure 4 ijms-23-14148-f004:**
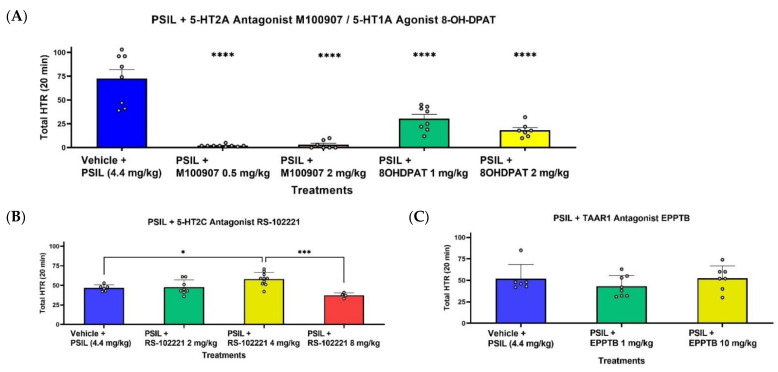
Total PSIL-induced HTR (**A**–**C**) during 20 min post injection with different co-treatments. (**A**) Effect of pre-treatment with M100907 0.5/2 mg/kg + PSIL 4.4 mg/kg, 8-OH-DPAT 1/2 mg/kg + PSIL 4.4 mg/kg or vehicle + PSIL 4.4 mg/kg (*n =* 7–8) (**B**) Effect of pre-treatment with RS-102221 2/4 8 mg/kg + PSIL 4.4 mg/kg or vehicle + PSIL 4.4 mg/kg (*n =* 4–11). (**C**) Effect of pre-treatment with EPPTB 1/10 mg/kg + PSIL 4.4 mg/kg or vehicle + PSIL 4.4 mg/kg (*n =* 6–8). Compared to vehicle (**A**,**C**) and between treatments (**B**), * *p* < 0.05, *** *p* < 0.001, **** *p* < 0.0001. Error bars represent SEM.

**Figure 5 ijms-23-14148-f005:**
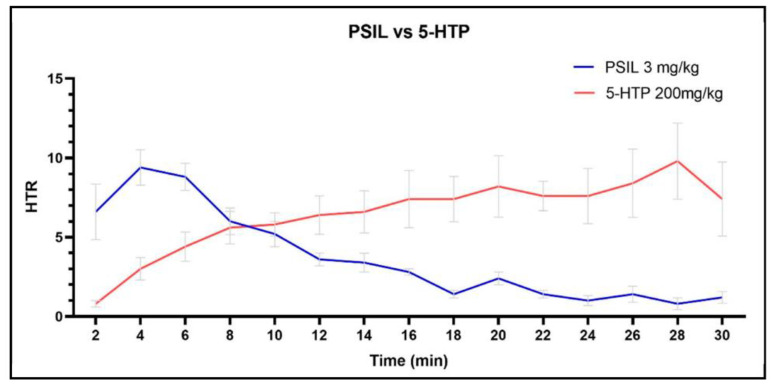
PSIL (3 mg/kg) and 5-HTP (200 mg/kg) HTR profile comparison based on maximal peak HTR. Error bars represent SEM.

## Data Availability

Qualified investigators may access data from this project by contacting the corresponding authors.

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
