# Peer review of "Role of 5-HT2A, 5-HT2C, 5-HT1A and TAAR1 Receptors in the Head Twitch Response Induced by 5-Hydroxytryptophan and Psilocybin: Translational Implications"

_ijms, 2022, doi:10.3390/ijms232214148_

Round 1
Reviewer 1 Report
In the current version of the manuscript, Shahar et al tested the effects of 5-HTP and psilocybin on HTR in C57BL/6J mice, as well as the pharmacological modulation of psilocybin-induced HTR by 5-HT2A antagonist M100907, the 5-HT1A agonist 8-OH-DPAT, the 5-HT2C antagonist RS10221 and the TAAR1 antagonist EPPTB.
Results show an inverted U-shaped dose-response effect of 5-HTP on HTR. At a single dose of 5-HTP, HTR is blocked by M100907, reduced by 8-OH-DPAT and EPPTB, and augmented by RS10221. Psilocybin also induced inverted U-shaped dose-response effect on HTR. At a single dose of psilocybin, HTR is blocked by M100907, reduced by 8-OH-DPAT, augmented by RS10221, and unaffected EPPTB.
General comment:
Generally speaking, the paper is descriptive and not mechanistic, and validates using a relatively new (although not novel) fully automated HTR system what others have previously shown with psychedelics (ie, 5-HT2A antagonists fully block HTR, and this behavior is modulated by 5-HT1A and 5-HT2C ligands). This reduces interest on the current version of the manuscript
More importantly, a recent publication testing a similar question (effect of pharmacological modulation of 5-HT1A and/or 5-HT2C receptor on psilocybin-induced HTR has just been published by a different group (PMID: 36049313). Of note, this recently published paper used translationally valid doses of psilocybin (0.125 o 3 mg/kg), opposite to (as already mentioned) the extremely high doses of psilocybin tested in this manuscript (more than 50 mg/kg).
More specific comments:
When an inhibitor and a psychedelic were tested, did mice receive the two adequate vehicles? This is not clearly mentioned
It is not clearly described when exactly the inhibitors were administered before psilocybin or 5-HTP
This reviewer could not download the supplementary material with the link provided in the current version of the manuscript. However, the number of mice for validation of the experimental system according to the methods for the HTR acquisition system is extremely low (n = 1-2 ) – this low statistical power also explains the “almost perfect” R2 = 1. This raises severe concerns about the validity of the method, and hence the conclusions of the study.
Description of Y axes is weak – HTR: HTR during how many minutes, total etc…
For the 5-HTP data, time courses last 30 min but it is very clear that mice are still showing HTR at 30 min (see Fig 1A for example). Longer time-courses are needed.
The number of HTR observed in different experiments but under the same experimental conditions is very different – For example in Fig 1B (200 mg/kg 5-HTP) HTR is about 100, but in Fig 2A (200 mg/kg 5-HTP) HTR is about 40, and ever lower in Fig 2C. This (again based on lack of controls to validate the HTR experimental system) raises concerns about the replicability of these findings.
Fig 2 lacks a vehicle/vehicle control group.
s
Post-hoc analyses are different among all the figures, or even within the same figure. Dunnett for Fig 1B, Turkey and Dunnett for Fig 2, Turkey for Fig 3, Turkey and Dunnett for Fig 4.
The physiological significance of the extremely high doses of psilocybin (51.2 mg/kg) is not discussed. Similarly, negative effects of these doses need to be tested (alterations in locomotor activity etc…)
Looks like some of the HTR are identical between “different” experimental sets. For example, vehicle+psilocybin 4.4 mg/kg group in Fig 4A and Fig 4B. If the mice are the same, the two groups should be analyzed and presented together as a single experiment.
Reviewer 2 Report
Dear Authors, dear editor,
Here is my review for the article “Role of 5-HT2A, 5-HT2C, 5-HT1A and TAAR1 receptors in the head twitch response induced by 5-hydroxytryptophan and psilocybin: Translational implications” submitted by Orr Shahar and collaborators for publication as research article in International Journal of Molecular Sciences.
Hallucinogens are now considered as potential treatments for depression and posttraumatic stress disorder, reviving the need to decipher their molecular mode of action.
The authors studied in mice model the responses to hallucinogens 5-hydroxytryptophan and psilocybin and evaluated by pharmacological inhibition their dependency to receptors 5-HT2A, 5-HT2C, 5-HT1A and TAAR1.
The study is carefully designed, driven and results are carefully interpretated.
Results clearly show that the two hallucinogens target much more receptors than the only classical 5HT2A receptor. Not much literature was available on the effect of 8-OH-DPAT and other 5-HT1A agonists on the response to psychedelic drugs.
The one and only phenotypical readout in the article is the classical head twitches. One originality of this work is that the authors used an automated magnetometer measurement with magnetic detectors attached to the mice ears. As mentioned in the manuscript, this method was initially described by Mario de la Fuente Revenga (J Neurosci Methods 2020, 334, 108595) but this is the first time I see the method used for such a large number of pharmacological conditions. Head twitch manual counting can be very time consuming or subjective and this study illustrates well what can be achieved by automatization.
I recommend the article for publication with only few minor suggestions/changes:
Correct M100907 in fig 2A
The authors should explain why their chose Psylocibin and 5-HTP amongst the collection of drugs targeting serotoninergic systems. What about DOI, mescaline, LSD or MDMA? Are they the most promising drugs for therapeutic applications ?
The authors mention another recently submitted paper where they evaluated the anti-obsessional effects of psylocibin by measuring marble burying (a screening test for) in male ICR mice (Singh et al, Submitted). Was the minimal dose of psylocibin for an observable phetnotypical response the same in both study. Some behavioural tests to hallucinogens might be mor sensitive than others.
Best regards
Round 2
Reviewer 1 Report
The authors have only partially addressed Reviewers’ comments
1- This article (PMID: 36049313) is directly related to the work presented here and a full paragraph in the discussion should be dedicated to compare similarities and differences between the two articles.
2- This Reviewer still thinks that the doses of psilocybin (51.2 mg/kg) are extremely high and additional controls should be included to address if mice misbehave just because of the dose (locomotor activity for example). Without these controls, the validity of the HTR assays is null.
3- Why a non-linear regression with Emax and EC50 values are presented in the response to the reviewers, but a more simplistic analysis (bar graph without non-linear regression) is included in the article?
4- Including exactly same experimental values from the same mice in different datasets is statistically incorrect, and should not be submitted for publication
5- This Reviewer requested for additional validation controls of the fully automated HTR system, but the authors have not satisfactorily addressed this concern. For example, in the methods section validation assays included experimental groups of n = 1. This is not correct from a statistical point of view.
Author Response
1- This article (PMID: 36049313) is directly related to the work presented here and a full paragraph in the discussion should be dedicated to compare similarities and differences between the two articles.
REPLY: We thank the Reviewer for this comment and have added such a paragraph in the Discussion. (Lines 315-347)
2- This Reviewer still thinks that the doses of psilocybin (51.2 mg/kg) are extremely high and additional controls should be included to address if mice misbehave just because of the dose (locomotor activity for example). Without these controls, the validity of the HTR assays is null.
REPLY: We accept the comment of the Reviewer and have removed the 51.2 mg/kg dose from all analyses and figures in the revised paper. Since the 51.2 mg/kg dose has been removed in accordance with the Reviewer’s suggestion, further controls suggested by the reviewer to establish the validity of the 51.2 mg/kg dose are no longer needed and have not been included.
3- Why a non-linear regression with Emax and EC50 values are presented in the response to the reviewers, but a more simplistic analysis (bar graph without non-linear regression) is included in the article?
REPLY: This analysis was included in the Reply to Reviewers that accompanied the previous revision of the paper. It was part of our response to Reviewer 1’s criticism of our use of a psilocybin dose in excess of 50 mg/kg. Since we have removed the higher dose (51.2 mg/kg), the additional analysis is no longer relevant. It was not included in the original manuscript but only in the Reply to Reviewer 1 and is not in the revised version.
4- Including exactly same experimental values from the same mice in different datasets is statistically incorrect, and should not be submitted for publication
REPLY: As suggested by the Reviewer, we have updated Figure 4A to combine the two separate modulators together with the shared control group. Therefore, experimental values from the same mice are no longer included in different datasets and the comment has been fully addressed.
5- This Reviewer requested for additional validation controls of the fully automated HTR system, but the authors have not satisfactorily addressed this concern. For example, in the methods section validation assays included experimental groups of n = 1. This is not correct from a statistical point of view.
REPLY: In this analysis we employed 5 mice. The correlation between HTRs measured manually and by magnetometer was examined across these 5 mice. We have added Supplementary Figure 2B which clearly shows the number of HTRs counted in each mouse by each method of measurement. There are 5 mice and two measurements of HTR for each mouse. We hope this clarifies what we have done and establishes that the experimental group was made up of 5 mice which we see as sufficient for this purpose. We respectfully contend that this statistical approach is legitimate and appropriate for examining the degree to which two methods of measurement resemble or differ from each other when measuring a behavioral variable such as HTR. Given that the correlation is as strong as it is, 5 mice are sufficient for this purpose. Furthermore, we note, as does the Reviewer, that the correlation between HTR measured manually and by magnetometer has been established by other authors and this matter is thus not a cardinal point. If the Reviewer insists we will remove this analysis from the manuscript so as not to further protract the review process but we believe this is not necessary and sincerely hope it will not be required.
